# Joint Task Offloading and Resource Allocation for Space–Air–Ground Collaborative Network

Chengli Mei [1], Cheng Gao [2], Heng Wang [1], Yanxia Xing [1], Ningyao Ju [2] and Bo Hu [2,*]

1 Chinatelecom Research Institute, Beijing 102209, China; meichl@chinatelecom.cn (C.M.); wangh26@chinatelecom.cn (H.W.); xingyx@chinatelecom.cn (Y.X.)
2 State Key Laboratory of Networking and Switching Technology, Beijing University of Posts and Telecommunications, Beijing 100876, China; gaoch@bupt.edu.cn (C.G.); juningyao@bupt.edu.cn (N.J.)
* Correspondence: hubo@bupt.edu.cn

**Abstract:** The space–air–ground collaborative network can provide computing service for ground users in remote areas by deploying edge servers on satellites and high-altitude platform (HAP) drones. However, with the growing number of ground devices required to be severed, it becomes imperative to address the issue of spectrum demand for the HAP drone to meet the access of a large number of users. In addition, the long propagation distance between devices and the HAP drone, and between the HAP drone and LEO satellites, will lead to high data transmission energy consumption. Motivated by these factors, we introduce a space–air–ground collaborative network that employs the non-orthogonal multiple access (NOMA) technique, enabling all ground devices to access the HAP drone. Therefore, all devices can share the same communication spectrum. Furthermore, the HAP drone can process part of the ground devices' tasks locally, and offload the rest to satellites within the visible range for processing. Based on this system, we formulate a weighted energy consumption minimization problem considering power control, computing frequency allocation, and task-offloading decision. The problem is solved by the proposed low-complexity iterative algorithm. Specifically, the original problem is decomposed into interconnected coupled subproblems using the block coordinate descent (BCD) method. The first subproblem is to optimize power control and computing frequency allocation, which is solved by a convex algorithm after a series of transformations. The second subproblem is to make an optimal task-offloading strategy, and we solve it using the concave–convex procedure (CCP)-based algorithm after penalty-based transformation on binary variables. Simulation results verify the convergence and performance of the proposed iterative algorithm compared with the two benchmark algorithms.

**Keywords:** space–ground–air collaborative network; mobile edge computing; drone communication; non-orthogonal multiple access (NOMA); task offloading and resource allocation



## 1. Introduction

With the rapid development of emerging applications such as Internet of Things (IOT) and Augmented Reality (AR)/Virtual Reality (VR), the fifth generation (5G) and future wireless networks need to meet the connectivity needs of massive mobile devices [1], while ensuring low latency [2] and low energy consumption [3]. In recent years, significant advancements have been made in low-earth orbit (LEO) satellite communication, with the successful commercialization of satellite constellations such as Starlink and OneWeb. Therefore, satellite communication is considered a crucial component of future networks. By carrying edge servers, LEO satellites can provide offloading services for ground devices (GDs) [4–6]. Although LEO satellites can directly provide services for ground devices, severe path loss causes ground devices to consume a lot of energy to upload data to LEO satellites, and satellite communication has a high communication delay. These have brought many new challenges to meeting the GDs' Quality of Service (QoS) requirements.

Recently, the high-altitude platform (HAP) drone has attracted the attention of many companies and researchers. Flying or hovering at an altitude of 20–50 km [7], the HAP drone is an unmanned aerial vehicle (UAV) in which a base station is deployed. HAP drones are mainly divided into solar stratospheric UAVs and floating air balloons. At present, companies such as Softbank, DLR, and Facebook are conducting in-depth research on HAP drones. The HAP drone can provide communication services by establishing line-of-sight (LoS) links with GD [8]. Furthermore, due to the payload of the HAP drone exceeding 100 kg [9,10], the HAP drone can also be equipped as an edge server to provide a task-offloading service for GDs. Compared with LEO satellite constellations such as Iridium and Starlink [11], the communication link between HAP drones and ground devices is more stable, and a ground device needs to consume less energy to transmit the same amount of data. Hence, space–air–ground collaborative networks are considered a promising approach to meeting the access and task-processing needs of massive ground devices.

However, when orthogonal multiple access (OMA) techniques such as frequency-division multiple access (FDMA) are used as the means of communication between the HAP drone and GDs, due to the large number of connected devices, the bandwidth available to each device is very limited [12]. Therefore, the non-orthogonal multiple access (NOMA) technique can be used for communication between the HAP drone and ground devices, and at this time, all devices can communicate with the HAP drone through the same spectrum [13]. The process of task offloading from ground devices to the HAP drone is the process of data uplink transmission. Using the NOMA technique in this process, all devices transmit data on the same spectrum, and the receiver (i.e., HAP drone) applies an advanced multi-user detection (MUD) technique such as successive interference cancellation (SIC) to extract data from different users based on their respective channel conditions [14].

Based on the above discussion, we propose a space–air–ground collaborative network to make the HAP drone and LEO satellites cooperate for mobile edge computing. Further, we formulate a weighted energy consumption minimization problem considering power control, computing frequency allocation, and task-offloading decision. The main challenges we face can be summarized as follows:

- How to properly control the transmission power of ground devices and the HAP drone? In this system, distinct communication methods are employed for the interactions between devices and the HAP drone, as well as between the HAP drone and LEO satellites. Consequently, it is imperative to develop specific power control strategies tailored to each transmission method.
- How to reasonably allocate the computing resources of each edge server? In this system, the HAP drone can directly obtain the computing capability of edge nodes (deployed on LEO satellites) and the computing requirements of GDs. Based on this information, HAP drones need to allocate a reasonable size of computing resources for each task.
- How to make a task-offloading decision? Considering a partial offloading model, each task can be split into two parts: the first part of it is processed at the HAP drone and the rest of it is offloaded to LEO satellite for processing. Therefore, the task-splitting strategy needs to be appropriately made. In addition, there are multiple LEO satellites in the field view of the HAP drone, and it is necessary to determine the target LEO satellite for each task offloading.

In this paper, we propose a space–air–ground collaborative network. Specifically, considering the difference in the number of access devices, the communication between the ground device and the HAP drone and the communication between the HAP drone and the LEO satellite adopt the NOMA technique and FDMA technique, respectively. Subsequently, the HAP drone can process all or part of the tasks locally, or offload device tasks to LEO satellites within its visible range. The main contributions of this paper are summarized as follows.

(1)  Considering the limited energy and resources of nodes in the system, we formulate an optimization problem of joint task offloading and resource allocation, aiming to minimize the weighted total energy consumption of the system. This problem is a mixed-integer non-linear programming (MINLP) problem.

(2)  We propose a low-complexity iterative algorithm based on a block coordinate descent (BCD) method to solve this MINLP problem, which reduces the complexity of the original problem by converting the original problem into two subproblems for the iterative solution. For the first subproblem, we transform the problem into a convex optimization problem and solve it with the convex algorithm. For the second subproblem, we convert this to a continuous variable problem by using a penalty-based transformation, and then we solve it by a concave–convex procedure (CCP)-based algorithm.

(3)  The simulation experiments have verified the convergence of the proposed algorithm in this paper. Furthermore, compared to the other two benchmark algorithms, the algorithm proposed in this paper consistently achieves a smaller overall system-weighted energy consumption under the same conditions.

The rest of this paper is structured as follows. The related works are discussed in Section 2. The system model is introduced in Section 3, including communication models, task offloading, and computation models. In Section 4, we formulate an energy minimization problem. Section 5 presents a proposed low-complexity iterative algorithm. The convergence and performance of the proposed algorithm are proven in Section 6. Finally, conclusions are given in Section 7.

## 2. Related Works

The related works of this paper include space–air–ground collaborative edge computing and NOMA-assisted edge computing. In the following, we introduce their specific research progress.

### 2.1. Space–Air–Ground Collaborative Edge Computing

Recently, several space–air–ground collaborative edge computings were introduced [9,15–19].

Nan Cheng et al. [15] presented a space–air–ground integrated network (SAGIN) edge/cloud computing architecture for offloading the computation-intensive applications considering remote energy and computation constraints, where flying UAVs provide near-user edge computing and satellites provide access to the cloud computing. In [16], the authors proposed a framework of edge computing-enabled SAGINs to support various Internet of Vehicles (EC-IoV) services for vehicles in remote areas whose main objective of the framework is to minimize the task completion time and satellite resource usage. A deep learning-driven offloading and caching algorithm is proposed to achieve real-time decision-making. In [17], Bomin Mao et al. considered the UAVs and satellites to offer wireless-powered IoT device edge computing and cloud computing services, respectively, and focus on the computation offloading problem and consider deep learning techniques to optimize the task success rate considering the energy dynamics and channel conditions. A deep learning-based optimization strategy for offloading policies is proposed, employing a long short-term memory (LSTM) model to effectively address the dynamic characteristics of energy harvesting performance.

The authors in [9] conducted a study on a satellite–air-integrated edge computing network to provide edge computing services for ground user (GUE) equipment by combining LEO satellites and HAPs. The authors minimized the weighted total energy consumption of GUEs, HAPs, and satellites in the network, including communication and calculation energy consumption, through joint GUE association, multi-user multi-input multi-output transmission precoding, calculation task allocation, and resource allocation. Ahmad Alsharoa and Mohamed-Slim Alouini [18] studied the goal of optimizing resource allocation and the location of HAP under the framework of the integration of ground base stations, high-altitude platforms, and satellite stations, and realized the improvement of the user

throughput. The authors divided the scene into two stages: the short-term stage and long-term stage to formulate and solve the optimization problem. In the short-term stage, user association and resource allocation are considered, and in the long-term stage, the location optimization problem of HAP is considered. Long Zhang et al. [19] proposed a satellite-to-air integrated computing (SAIC) architecture in a disaster environment, in which the computing tasks from two layers of users (i.e., ground/air user equipment) were either executed locally on HAPs or offloaded to LEO satellites for computing. Under SAIC architecture, the problem of joint two-layer user association and unloading decisions with the goal of maximizing the total rate is studied.

### 2.2. NOMA-Assisted Edge Computing

The authors in [20] integrated cloud-edge computing and NOMA to propose a network communication model, which can provide users with energy-efficient and low-latency services. The model considers the energy consumption, transmission delay, and quality of service; the authors jointly optimized the offloading decision and its radio resource allocations for NOMA transmission to reduce the system cost (the weighted sum of consumed energy and delay). Zhiguo Ding et al. [21] proposed a hybrid NOMA-MEC scheme, in which a user first offloads parts of its task by using a time slot allocated to another user and then offloads the remainder of its task during a time slot solely occupied by itself, where the power and time allocation is jointly optimized to reduce the energy consumption of computation offloading. In [22], the authors investigated the edge user allocation problem in the NOMA-based MEC system. The authors introduced a decentralized game-theoretic approach to allocating maximum users to edge servers in a specific area at the lowest computing resource and transmit power costs. The authors in [23] proposed a novel cooperative MEC that exploits the combination of NOMA and multiple helpers. In the proposed system featuring a user, multiple helpers, and a base station, the user can simultaneously offload its computation-intensive tasks to the helpers using NOMA when there is no strong direct transmission link between the user and the BS. Then, the helpers can compute and offload these tasks through NOMA. Ming Zeng et al. [24] aimed to minimize the overall delay for offloading in a multi-user NOMA-MEC network under maximum power constraint and maximum energy constraint for offloading users, and they proposed a NOMA scheme that can achieve substantial delay reduction compared with time division multiple access (TDMA). In [25], a NOMA-based vehicle edge computing (VEC) network model is proposed, and the cost minimization problem is constructed. Under the premise of ensuring the delay tolerance of all vehicle users (VUEs), the total system cost is minimized through the joint optimization of offloading decision-making, VUE clustering, subchannel and computation resource allocation, and transmission power control. The authors in [26] proposed a general hybrid NOMA-MEC offloading strategy, which includes conventional orthogonal multiple access (OMA) and pure NOMA-based offloading as special cases. A multi-objective optimization problem is formulated to minimize the energy consumption for MEC offloading.

### 3. System Model

In the space–air–ground collaborative network, there are multiple LEO satellites equipped with mobile edge computing servers within the visual range of the HAP drone, which can be denoted as $\mathcal{M} = \{1, 2, \ldots, M\}$. All satellites can provide edge computing services for ground devices (GDs). The computation capacity of LEO satellite $m$ is $F_m$; this means that the maximum number of CPU cycles per second for satellite $m$ is $F_m$. In this scenario, the HAP drone is also equipped with an edge computing server. The computation capacity of the HAP drone is $F_h$. On the ground, there are $N$ GDs, which can be denoted as $\mathcal{N} = \{1, 2, \ldots, N\}$. For GD $n$, its task can be denoted as $\{D_n, c_n, T_n\}$, where $D_n$ is the input data size of task $n$, $c_n$ represents the number of CPU cycles required to process 1bit task $n$, and $T_n$ represents the maximum delay to process the task $n$. The computation capacity of GD $n$ can be denoted as $F_n$.

*3.1. NOMA-Based Communication Model*

3.1.1. GD-HAP Drone Uplink Communication Model

All GDs transmit data to the HAP drone based on the NOMA technique. The received signal of the HAP drone from GD $n$ can be denoted as

$$y_n = \underbrace{h_{h,n}\sqrt{p_n}s_n}_{desired\ signal} + \underbrace{\sum_{i\in\mathcal{N}\setminus\{n\}} h_{h,i}\sqrt{p_i}s_i}_{intra-interference} + \underbrace{\hat{n}_n}_{noise} \tag{1}$$

where $h_{h,n}$, $h_{h,i}$ are channel gains between the HAP drone and GD $n$, and between the HAP drone and GD $i$. $p_n$, $p_i$ are transmission power of GD $n$ and GD $i$. $\hat{n}_n$ is the additional white Gaussian noise (AWGN), which is considered to satisfy the distribution of $\hat{n} \sim \boldsymbol{CN}(0,\sigma^2)$.

The signal-to-interference-plus-noise-ratio (SINR) at HAP drone from GD $n$ is

$$\gamma_n = \frac{|h_{h,n}|^2 p_n}{\sum_{i\in\mathcal{N}\setminus\{n\}} |h_{h,i}|^2 p_i + \sigma^2} \tag{2}$$

Then, we can get the data rate between HAP and GD $n$

$$R_n = B_h \log_2(1 + \gamma_n) \tag{3}$$

where $B_h$ is the bandwidth for each GD.

3.1.2. Consideration of SIC Decoding

In the stage of device upload data, all GDs transmit their tasks to the HAP drone simultaneously based on the NOMA technique. All GDs are sorted by channel gains

$$|h_{h,1}| \geq |h_{h,2}| \geq \dots \geq |h_{h,N}| \tag{4}$$

Then, the HAP drone utilizes the SIC technique to decode data from GDs. According to the principles of SIC, the HAP drone first decodes the information from the GD with larger channel gain, and then removes it from the interference terms of other GDs. Therefore, the offloading rate of $i$-th GD can be expressed as [27]

$$R_{i,h} = B_h \log_2(1 + \frac{|h_{h,i}|^2 p_i}{\sum_{j=i+1}^{N} |h_{h,j}|^2 p_j + \sigma^2}) \tag{5}$$

*3.2. FDMA-Based Communication Model*

The communication between the HAP drone and LEO satellites adopts the Frequency Division Multiple Access (FDMA) technique; the data rate between the HAP drone and LEO satellite $m$ can be denoted as

$$R_{h,m} = B_s \log_2(1 + \frac{|h_{h,m}|^2 p_{h,m}}{\sigma^2}) \tag{6}$$

where $h_{h,m}$ is the channel gain between the HAP drone and LEO satellite $m$. $B_s$ is the allocated bandwidth for each LEO satellite. Assuming that the total available bandwidth is $B_{total}$, the bandwidth allocated to each LEO is $B_s = \frac{B_{total}}{M}$.

Based on 3GPP specifications, the free space path loss (FSPL) in dB between GD and HAP drone, and between HAP drone and LEO satellite can be expressed as [28].

$$FSPL(d_{h,i}, fc) = 32.45 + 20\log_{10}(f_c) + 20\log_{10}(d_{h,i}) \tag{7}$$

where $d_{h,i} = \sqrt{(x_h - x_i)^2 + (y_h - y_i)^2 + (h_h - h_i)^2}$ is the distance between the HAP drone and LEO satellite, or between the HAP drone and GD. $\{x_h, y_h, h_h\}$ is the coordinate of the

HAP drone. $\{x_i, y_i, h_i\}$ is the coordinate of the LEO satellite or GD $i$ ($i \in \mathcal{M} \cup \mathcal{N}$). $f_c$ is the carrier frequency in GHz of the transmitted signal. Therefore, the channel gain between HAP and LEO or GD $i \in \{\mathcal{M} \cup \mathcal{N}\}$ can be formulated as

$$\left| h_{h,i} \right|^2 = 10^{-\frac{FSPL(d_{h,i}, fc)}{10}} \tag{8}$$

*3.3. Task Offloading and Computation Model*

3.3.1. GD-HAP Drone Task Offloading Model

In this system, all GDs offload their tasks to the HAP drone. For GD $n$, its task transmission time from GD $n$ to the HAP drone can be formulated as

$$t_n^{trans} = \frac{D_n}{R_{n,h}} \tag{9}$$

And the energy consumption of GD $n$ to offload its task to HAP drone can be expressed as

$$E_n^{trans} = p_n \times t_n^{trans} = p_n \frac{D_n}{R_{n,h}} \tag{10}$$

3.3.2. HAP Drone Transmission and Computation Model

In this paper, we adopt a partial offloading protocol [9]. For GD $n$, the HAP and the LEO satellite process different portions of its computation task. When the HAP drone receives the task of GD, it can execute part of it on the local server. At the same time, the HAP drone offloads the rest of the task to the LEO satellite, which is executed by the LEO satellite server. GD $n$'s task can be divided into two parts: $\delta_n$ ($0 \leq \delta_n \leq D_n$) bits are executed at the HAP drone's MEC server, and $D_n - \delta_n$ bits are offloaded to the LEO satellite for processing. Therefore, the time delay for executing GD $n$'s task at HAP can be formulated as

$$t_{n,h} = \frac{c_n \delta_n}{f_{n,h}} \tag{11}$$

And the energy consumption for executing GD $n$'s task at HAP can be expressed as

$$E_{n,h} = \kappa_h (f_{n,h})^2 c_n \delta_n \tag{12}$$

where $f_{n,h}$ is the computation resource allocated to GD $n$'s task by the HAP drone. $\kappa_h$ is a constant relative to the hardware architecture of the HAP drone.

When the HAP drone offloads the rest of the GD $n$'s task to the LEO satellite $m$, the transmission delay of task offloading can be denoted as

$$t_{n,h,m} = \alpha_{n,m} \frac{D_n - \delta_n}{R_{h,m}} + T_{h,m} \tag{13}$$

where $\alpha_{n,m} \in \{0,1\}, \forall n \in \mathcal{N}, m \in \mathcal{M}$ is the offloading indicator of GD $n$ by the HAP drone. $\alpha_{n,m} = 1$ indicates that the HAP drone offloads GD $n$'s task to the LEO satellite $m$, and $\alpha_{n,m} = 0$, otherwise. $T_{h,m}$ is the round-trip propagation delay between the HAP drone and LEO satellite $m$, which can be formulated as

$$T_{h,m} = 2 \frac{\sqrt{(x_h - x_m)^2 + (y_h - y_m)^2 + (h_h - h_m)^2}}{c} \tag{14}$$

where $c$ is the speed of light. The energy consumption of task offloading can be expressed as

$$E_{n,h,m} = p_{h_m} * t_{n,h,m} = p_{h,m} (\alpha_{n,m} \frac{D_n - \delta_n}{R_{h,m}} + T_{h,m}) \tag{15}$$

### 3.3.3. LEO Satellite Computation Model

In this paper, the LEO cannot offload the GD's task to another LEO satellite. If the HAP drone offloads GD $n$'s task to LEO satellite $m$, the computation delay of task $n$ executed on LEO $m$ can be denoted as

$$t_{n,m} = \frac{\alpha_{n,m} c_n (D_n - \delta_n)}{f_{n,m}} \tag{16}$$

where $f_{n,m}$ is the computation resource allocated to GD $n$'s task by LEO satellite $m$. Furthermore, the energy consumption of task computation on LEO satellite $m$ can be formulated as

$$E_{n,m} = \alpha_{n,m} \kappa_m (f_{n,m})^2 c_n (D_n - \delta_n) \tag{17}$$

where $\kappa_m$ is a constant relative to the hardware architecture of LEO satellite $m$.

### 3.4. Overall Delay and Energy Consumption

The processing delay of GD $n$'s task can be divided into two parts. The first part is the time delay processed by the HAP drone, and the second part is the time delay processed by the LEO satellite, which can be expressed as

$$T_{n,h}^{all} = \frac{D_n}{R_{n,h}} + \frac{c_n \delta_n}{f_{n,h}} \tag{18}$$

and

$$
\begin{aligned}
T_{n,m,s}^{all} &= t_n^{trans} + t_{n,h,m} + t_{n,m} \\
&= \frac{D_n}{R_{n,h}} + \alpha_{n,m} \frac{D_n - \delta_n}{R_{h,m}} + T_{h,m} + \frac{\alpha_{n,m} c_n (D_n - \delta_n)}{f_{n,m}}
\end{aligned} \tag{19}
$$

The weighted energy consumption of the system can be formulated as

$$
\begin{aligned}
E_{sys} &= \omega_g E^{trans} + \omega_h E_h + \omega_s E_s \\
&= \omega_g \sum_{n=1}^{N} \left( p_n \frac{D_n}{R_{n,h}} \right) + \omega_h \sum_{n=1}^{N} \kappa_h (f_{n,h})^2 c_n \delta_n \\
&\quad + \omega_h \sum_{n=1}^{N} \sum_{m=1}^{M} \left( p_{h,m} \alpha_{n,m} \left( \frac{D_n - \delta_n}{R_{h,m}} + T_{h,m} \right) \right) \\
&\quad + \omega_s \sum_{n=1}^{N} \sum_{m=1}^{M} \left( \alpha_{n,m} \kappa_m (f_{n,m})^2 c_n (D_n - \delta_n) \right)
\end{aligned} \tag{20}
$$

## 4. Strategy Design and Problem Formulation

In this section, we first present the process of resource allocation and task offloading. Then, we formulate the optimization problem of joint task offloading and resource allocation to minimize the weight energy consumption of the system.

### 4.1. Strategy Design

This paper studies the space–air–ground collaborative network shown in Figure 1, where the HAP drone directly connects with LEO satellites and ground devices through Ka-band and C-band, respectively. The HAP drone serves as the control node in this system, responsible for collecting information from all nodes in the system (user task information, satellite computing resource information, etc.), and making and distributing task offloading and resource allocation strategy. The implementation process of task offloading and resource allocation strategy design can be divided into four steps:

- Information collection: in this step, the HAP drone collects information from LEO satellites $\mathcal{M}$ within its visual range, and information from GDs $\mathcal{N}$ connected to the HAP drone (including computational resources, channel information, etc.).
- Task-offloading request: In this step, the GDs connected to the HAP drone send a task-offloading request to the HAP drone, which includes specific information about the task, such as the data size, required CPU cycles per bit of data processing, and the maximum processing tolerance delay.
- Strategy-making and distribution: After the HAP drone collects information from each node and receives task-offloading requests from the GDs, the HAP drone makes an appropriate strategy for resource allocation and task offloading based on this information. The resource allocation and task-offloading strategy will be sent to the respective GDs and LEO satellites via C-band and Ka-band.
- Task processing: After receiving the resource allocation and task-offloading strategy from the HAP drone, the GDs send the task to the HAP drone according to the strategy, and then the HAP drone and LEO satellites process the tasks based on the resource allocation and task-offloading strategy.

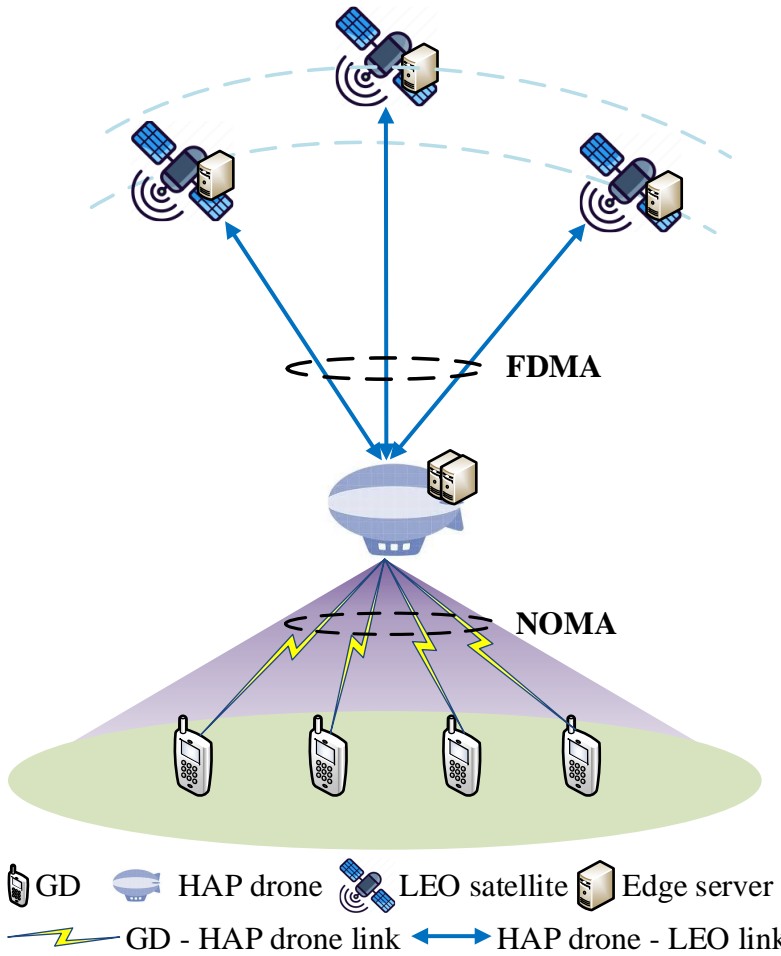

**Figure 1.** The scenario of the space–air–ground collaborative network.

*4.2. Problem Formulation*

In the previous section, we defined the system's weighted energy consumption as the weighted sum of the energy consumption of each node. In order to minimize the system's

weighted energy consumption, a joint optimization problem of task offloading and resource allocation is formulated as follows:

$$\mathcal{OP} : \min_{\boldsymbol{P},\boldsymbol{F},\boldsymbol{\alpha},\boldsymbol{\delta}} E_{sys} \tag{21a}$$

$$\text{s.t.} \quad 0 \leq p_n \leq P_n^{max}, \forall n \in \mathcal{N} \tag{21b}$$

$$0 \leq p_{h,m} \leq P_h^{max}, \forall m \in \mathcal{M} \tag{21c}$$

$$0 \leq f_{n,h}, f_{n,m}, \forall n \in \mathcal{N}, m \in \mathcal{M} \tag{21d}$$

$$\sum_{n=1}^{N} f_{n,h} \leq F_h \tag{21e}$$

$$\sum_{n=1}^{N} f_{n,m} \leq F_m, \forall m \in \mathcal{M} \tag{21f}$$

$$0 \leq \delta_n \leq D_n, \forall n \in \mathcal{N} \tag{21g}$$

$$T_{n,h}^{all} \leq T_n, \forall n \in \mathcal{N} \tag{21h}$$

$$T_{n,m,s}^{all} \leq T_n, \forall n \in \mathcal{N}, m \in \mathcal{M} \tag{21i}$$

$$\alpha_{n,m} \in \{0,1\}, \forall n \in \mathcal{N}, m \in \mathcal{M} \tag{21j}$$

$$\sum_{m=1}^{M} \alpha_{n,m} = 1, \forall n \in \mathcal{N} \tag{21k}$$

where $\boldsymbol{P} = \{p_n | \forall n \in \mathcal{N}\} \cup \{p_{h,m} | \forall m \in \mathcal{M}\} \in \mathbb{Z}^{1 \times (M+N)}$ is the set of all transmit powers, $\boldsymbol{F} \in \mathbb{Z}^{N \times (M+1)}$ is the set of total computation resources for the HAP drone and all LEO satellites, $\boldsymbol{\alpha} = \{\alpha_{n,m} | \forall n \in \mathcal{N}, m \in \mathcal{M}\} \in \mathbb{Z}^{N \times M}$ and $\boldsymbol{\delta} = \{\delta_n | \forall n \in \mathcal{N}\} \in \mathbb{Z}^{1 \times N}$ are collections of target access LEO satellite selection and task-splitting decisions.

Constraints (21b) and (21c) indicate that the transmit power cannot exceed the maximum power. Constraint (21d) represents that the CPU frequency allocation variables are non-negative. Constraints (21e) and (21f) are constraints of the total computation capacity for the HAP drone and each LEO satellite. Constraint (21g) is the constraint of task-splitting variables. Constraints (21h) and (21i) ensure that the processing delay of the task cannot exceed the maximum tolerable delay. Constraints (21j) and (21k) restrict the variables $\alpha_{n,m}$ to binary integer variables, and each task cannot be offloaded to multiple LEO satellites.

## 5. Algorithm Design for $\mathcal{OP}$

In this section, we propose a joint task-offloading and resource allocation optimization scheme to solve the problem $\mathcal{OP}$. First, we decouple the $\mathcal{OP}$ to two subproblems based on the BCD method, one for the optimization of all devices' transmission power and computation resources with fixed task-offloading decisions $\{\boldsymbol{\alpha}, \boldsymbol{\delta}\}$, which can be denoted as

$$\mathcal{P}1 : \min_{\boldsymbol{P},\boldsymbol{F}} E_{sys} \tag{22a}$$

$$\text{s.t.} \quad (21b)–(21f), (21h), (21i) \tag{22b}$$

Furthermore, task-offloading decisions are optimized based on fixed transmission and computation allocation strategy, and this subproblem can be denoted as

$$\mathcal{P}2 : \min_{\boldsymbol{\alpha},\boldsymbol{\delta}} E_{sys} \tag{23a}$$

$$\text{s.t.} \quad (21g)–(21k) \tag{23b}$$

By alternately solving these two subproblems, we can obtain an optimized resource allocation and task-offloading strategy.

### 5.1. Algorithm Design for $\mathcal{P}1$

Problem $\mathcal{P}1$ is non-convex and thus difficult to solve directly. To solve the problem $\mathcal{P}1$, in this subsection, we convert this to convex form and solve it by convex algorithm. Considering that the transmit power allocation problem of the HAP drone is non-convex, we can denote

$$
\begin{aligned}
f_1(\boldsymbol{P_h}) &= \min_{\boldsymbol{P_h}} \sum_{n=1}^{N} \sum_{m=1}^{M} (p_{h,m} \alpha_{n,m} (\frac{D_n - \delta_n}{R_{h,m}} + T_{h,m})) \\
&= \min_{\boldsymbol{P_h}} \sum_{m=1}^{M} (\sum_{n=1}^{N} \alpha_{n,m} (D_n - \delta_n)) \frac{p_{h,m}}{B_s \log_2(1 + \frac{p_{h,m} |h_{h,m}|^2}{\sigma^2})} \\
&\quad + \sum_{m=1}^{M} (\sum_{n=1}^{N} \alpha_{n,m}) T_{h,m} p_{h,m}
\end{aligned}
\tag{24a}
$$

$$
\text{s.t.} \quad (21d) \tag{24b}
$$

where $\boldsymbol{P_h} = \{p_{h,m} | m \in \mathcal{M}\} \in \mathbb{Z}^{1 \times M}$, which is the set of transmit power from the HAP drone to LEO satellites. We introduce new variables $\boldsymbol{\tau_h} = \{\tau_{h,m} | m \in \mathcal{M}\} \in \mathbb{Z}^{1 \times M}$, which can be denoted as

$$
\tau_{h,m} = \frac{1}{R_{h,m}} = \frac{1}{B_s \log_2(1 + \frac{p_{h,m} |h_{h,m}|^2}{\sigma^2})}, \forall m \in \mathcal{M} \tag{25}
$$

then, $\boldsymbol{P_h}$ can be expressed as

$$
p_{h,m} = \frac{\sigma^2}{|h_{h,m}|^2} (2^{\frac{1}{B_s \tau_{h,m}}} - 1), \forall m \in \mathcal{M} \tag{26}
$$

We can rewrite $f_1(\boldsymbol{P_h})$ as

$$
\begin{aligned}
g_1(\boldsymbol{\tau_h}) &= \min_{\boldsymbol{\tau_h}} \sum_{m=1}^{M} (\sum_{n=1}^{N} \alpha_{n,m} (D_n - \delta_n)) \frac{\sigma^2}{|h_{h,m}|^2} (2^{\frac{1}{B_s \tau_{h,m}}} - 1) \tau_{h,m} \\
&\quad + \sum_{m=1}^{M} (\sum_{n=1}^{N} \alpha_{n,m}) T_{h,m} \frac{\sigma^2}{|h_{h,m}|^2} (2^{\frac{1}{B_s \tau_{h,m}}} - 1)
\end{aligned}
\tag{27a}
$$

$$
\text{s.t.} \quad \frac{1}{B_s \log_2(1 + \frac{p_h^{max} |h_{h,m}|^2}{\sigma^2})} \leq \tau_{h,m} \forall m \in \mathcal{M} \tag{27b}
$$

This is a convex optimization problem, which is easy to solve. Further, considering the transmit power allocation problem of all GDs is also non-convex. Let $\boldsymbol{P_n} = \{p_n | \forall n \in \mathcal{N}\} \in \mathbb{Z}^{1 \times N}$ be the set of all ground devices' transmit power, and we can denote the GD's transmit power allocation problem as

$$
f_2(\boldsymbol{P_n}) = \min_{\boldsymbol{P_n}} \sum_{n=1}^{N} p_n \frac{D_n}{B_h \log_2(1 + \frac{|h_{h,n}|^2 p_n}{\sum_{j=n+1}^{N} |h_{h,j}|^2 p_j + \sigma^2})} \tag{28a}
$$

$$
\text{s.t.} \quad (21c) \tag{28b}
$$

We introduce new variables $\{t_{1,n}|\forall n \in \mathcal{N}\} \in \mathbb{Z}^{1 \times N}$, which represent the transmission delay for GDs to transmit the task data to the HAP. Furthermore, we can transform the $f_2(\boldsymbol{P_n})$ to

$$g_2(\boldsymbol{P_n}, \{t_{1,n}\}) = \min_{\boldsymbol{P_n}, \{t_{1,n}\}} \sum_{n=1}^{N} p_n t_{1,n} \tag{29a}$$

$$\text{s.t.} \quad \frac{D_n}{B_h \log_2(1 + \frac{|h_{h,n}|^2 p_n}{\sum_{j=n+1}^{N}|h_{h,j}|^2 p_j + \sigma^2})} \le t_{1,n}, \forall n \in \mathcal{N} \tag{29b}$$

$$(21c) \tag{29c}$$

Note that the constraint (29b) is non-convex, which can be rewritten as

$$D_n \le t_{1,n} B_h \log_2(1 + \frac{|h_{h,n}|^2 p_n}{\sum_{j=n+1}^{N}|h_{h,j}|^2 p_j + \sigma^2})$$
$$= t_{1,n} B_h \log_2(\sum_{j=n}^{N}|h_{h,n}|^2 p_j + \sigma^2) - t_{1,n} B_h \log_2(\sum_{j=n+1}^{N}|h_{h,n}|^2 p_j + \sigma^2) \tag{30}$$

To solve non-convex constraint, we introduce new variables $t_{2,n} \le B_h \log_2(\sum_{j=n}^{N}|h_{h,n}|^2 p_j + \sigma^2)$ and $t_{3,n} \ge B_h \log_2(\sum_{j=n+1}^{N}|h_{h,n}|^2 p_j + \sigma^2)$. Thus, constraint (30) can be rewritten as $D_n \le t_{1,n} t_{2,n} - t_{1,n} t_{3,n}, \forall n \in \mathcal{N}$. It is obvious that this is also non-convex. We can transform it into the Difference of Convex (DC) program

$$0 \ge D_n - t_{1,n} t_{2,n} + t_{1,n} t_{3,n}$$
$$= D_n + \frac{t_{1,n}^2 + t_{2,n}^2}{2} - \frac{(t_{1,n} + t_{2,n})^2}{2} + \frac{(t_{1,n} + t_{3,n})^2}{2} - \frac{t_{1,n}^2 + t_{3,n}^2}{2} \tag{31}$$

Further, we transform the above formula into a convex optimization form using the Taylor expansion around current point $\{t'_{1,n}, t'_{2,n}, t'_{3,n}|\forall n \in \mathcal{N}\}$.

$$0 \ge D_n + \frac{t_{1,n}^2 + t_{2,n}^2}{2} - \frac{(t'_{1,n} + t'_{2,n})^2}{2} - (t'_{1,n} + t'_{2,n})(t_{1,n} - t'_{1,n} + t_{2,n} - t'_{2,n})$$
$$+ \frac{(t_{1,n} + t_{3,n})^2}{2} - \frac{(t'_{1,n})^2 + (t'_{3,n})^2}{2} - t'_{1,n}(t_{1,n} - t'_{1,n}) - t'_{3,n}(t_{3,n} - t'_{3,n}) \tag{32}$$

Now, this constraint is convex. Furthermore, the $E_{sys}$ can be rewritten as

$$\hat{E}_{sys} = \omega_g \sum_{n=1}^{N} p_n t_{1,n} + \omega_h \sum_{n=1}^{N} \kappa_h (f_{n,h})^2 c_n \delta_n$$
$$+ \omega_h \sum_{m=1}^{M} (\sum_{n=1}^{N} \alpha_{n,m}(D_n - \delta_n)) \frac{\sigma^2}{|h_{h,m}|^2} (2^{\frac{1}{B_s \tau_{h,m}}} - 1) \tau_{h,m}$$
$$+ \omega_h \sum_{m=1}^{M} (\sum_{n=1}^{N} \alpha_{n,m}) T_{h,m} \frac{\sigma^2}{|h_{h,m}|^2} (2^{\frac{1}{B_s \tau_{h,m}}} - 1)$$
$$+ \omega_s \sum_{n=1}^{N} \sum_{m=1}^{M} (\alpha_{n,m} \kappa_m (f_{n,m})^2 c_n (D_n - \delta_n)) \tag{33}$$

Through the above transformation of problem $\mathcal{P}\mathbf{1}$, we can rewrite $\mathcal{P}\mathbf{1}$ as problem $\mathcal{P}\mathbf{3}$, and solving problem P3 can realize the solution of problem $\mathcal{P}\mathbf{1}$. $\mathcal{P}\mathbf{3}$ can be formulated as

$$\mathcal{P}\mathbf{3}: \quad \min_{\mathbf{F},\tau_h,\mathbf{t}} \hat{E}_{sys} \tag{34a}$$

$$\text{s.t.} \quad \frac{1}{B_s \log_2(1 + \frac{p_h^{max}|h_{h,m}|^2}{\sigma^2})} \leq \tau_{h,m} \forall m \in \mathcal{M} \tag{34b}$$

$$B_h \log_2(\sum_{j=n}^{N}|h_{h,n}|^2 p_j + \sigma^2) \geq t_{2,n}, \forall n \in \mathcal{N} \tag{34c}$$

$$B_h \log_2(\sum_{j=n+1}^{N}|h_{h,n}|^2 p_j + \sigma^2) \leq t_{3,n}, \forall n \in \mathcal{N} \tag{34d}$$

$$t_{1,n} + \frac{c_n \delta_n}{f_{n,h}} \leq T_n, \forall n \in \mathcal{N} \tag{34e}$$

$$t_{1,n} + \alpha_{n,m}(D_n - \delta_n)\tau_{h,m} + T_{h,m}$$
$$+ \frac{\alpha_{n,m}c_n(D_n - \delta_n)}{f_{n,m}} \leq T_n, \forall n \in \mathcal{N} \tag{34f}$$

$$(21b), (21d) - (21f), (32) \tag{34g}$$

where $\mathbf{t} = \{t_{1,n}, t_{2,n}, t_{3,n} | n \in \mathcal{N}\}$. This is a convex optimization problem, and we can solve it by using existing convex solvers, e.g., CVX toolbox [29].

### 5.2. Algorithm Design for $\mathcal{P}\mathbf{2}$

With fixed $\{\mathbf{P}, \mathbf{F}\}$, the optimization objective $E_{sys}$ is only related to $E_h$ and $E_s$. This also means that $E^{trans}$ in the objective function does not need to be optimized in this problem, so the $\mathcal{P}\mathbf{2}$ can be rewritten as

$$\mathcal{P}\mathbf{4}: \quad \min_{\alpha,\delta}(\omega_h E_h + \omega_s E_s) \tag{35a}$$

$$\text{s.t.} \quad (21g)–(21k) \tag{35b}$$

The constraint (21j) shows that $\alpha$ are 0–1 integer variables in $\mathcal{P}\mathbf{4}$, so this is an integer programming problem. The objective function (35a) and constraint (21i) are non-convex. This is an MINLP problem, which is difficult to solve. To solve this problem, we introduce the auxiliary variables $\grave{\alpha} = \{\grave{\alpha}_{n,m} | \forall n \in \mathcal{N}, m \in \mathcal{M}\} \in \mathbb{Z}^{N \times M}$; the constraint $(21j)$ can be transformed to [30]

$$\alpha_{n,m} * (1 - \grave{\alpha}_{n,m}) = 0, \forall n \in \mathcal{N}, m \in \mathcal{M} \tag{36}$$

and

$$\alpha_{n,m} = \grave{\alpha}_{n,m}, \forall n \in \mathcal{N}, m \in \mathcal{M} \tag{37}$$

To simplify the solution to the $\mathcal{P}\mathbf{4}$, we can add the constraints (36) and (37) as penalties to the objective function of $\mathcal{P}\mathbf{4}$, at which point $\mathcal{P}\mathbf{4}$ can be rewritten as problem $\mathcal{P}\mathbf{5}$

$$\mathcal{P}\mathbf{5}: \quad \min_{\alpha,\delta}(\omega_h E_h + \omega_s E_s) + \lambda \sum_{n=1}^{N}\sum_{m=1}^{M}(|(\alpha_{n,m} - \grave{\alpha}_{n,m})|^2 + |\alpha_{n,m}(1 - \grave{\alpha}_{n,m})|^2) \tag{38a}$$

$$\text{s.t.} \quad 0 \leq \alpha_{n,m} \leq 1, \forall n \in \mathcal{N}, m \in \mathcal{M} \tag{38b}$$

$$(21g)–(21i), (21k) \tag{38c}$$

Note that $\mathcal{P}\mathbf{5}$ is still non-convex because (35a) and (21i) are non-convex. To simplify the problem-solving process, we begin by transforming the problem into a Difference of Convex (DC) program problem. Based on the CCP method, we can find a non-convex feasible set near the current feasible point by the iterative convex approximation method and then solve a new convex approximation in each iteration [31]. We convert the non-

convex problem into a convex optimization problem by performing Taylor expansion on the current point, and $(\omega_h E_h + \omega_s E_s)$ can be rewritten as

$$
\begin{aligned}
E' = &\ \omega_h \sum_{n=1}^{N} \kappa_h (f_{n,h})^2 c_n \delta_n \\
&+ \omega_h \sum_{n=1}^{N} \sum_{m=1}^{M} p_{h,m} \Big( \frac{D_n}{R_{h,m}} + T_{h,m} \Big) \alpha_{n,m} \\
&+ \omega_h \sum_{n=1}^{N} \sum_{m=1}^{M} \frac{p_{h,m}}{R_{h,m}} \Big( \frac{1}{2} (\alpha_{n,m}^2 + \delta_n^2) - \frac{(\alpha'_{n,m} + \delta'_n)^2}{2} \\
&\quad - ((\alpha_{n,m} - \alpha'_{n,m}))((\alpha'_{n,m} + \delta'_n) - ((\delta_n - \delta'_n)((\alpha'_{n,m} + \delta'_n)) \\
&+ \omega_s \sum_{n=1}^{N} \sum_{m=1}^{M} \kappa_m (f_{n,m})^2 c_n D_n \alpha_{n,m} \\
&- \omega_s \sum_{n=1}^{N} \sum_{m=1}^{M} \kappa_m (f_{n,m})^2 c_n \Big( \frac{1}{2} (\alpha_{n,m}^2 + \delta_n^2) - \frac{(\alpha'_{n,m} + \delta'_n)^2}{2} \\
&\quad - ((\alpha_{n,m} - \alpha'_{n,m}))((\alpha'_{n,m} + \delta'_n) - ((\delta_n - \delta'_n)((\alpha'_{n,m} + \delta'_n))
\end{aligned}
\tag{39}
$$

where $\{\alpha'_{n,m} | \forall n \in \mathcal{N}, m \in \mathcal{M}\}$ and $\{\delta'_n | \forall n \in \mathcal{N}\}$ denote the current feasible point. Furthermore, (21i) can be rewritten as

$$
\Big( \frac{D_n}{R_{n,h}} + \frac{c_n D_n}{f_{n,m}} \Big) \alpha_{n,m} + \Big( \frac{D_n}{R_{n,h}} + \frac{c_n}{f_{n,m}} \Big) \theta_{n,m} + T_{h,m} + \frac{D_n}{R_{n,h}} \le T_n, \forall n \in \mathcal{N}, m \in \mathcal{M}
\tag{40}
$$

where $\theta_{n,m} \ge \frac{\alpha_{n,m}^2 + \delta_n^2}{2} - \frac{(\alpha'_{n,m} + \delta'_n)^2}{2} - ((\alpha'_{n,m} + \delta'_n))(\alpha_{n,m} - \alpha'_{n,m} + \delta_n - \delta'_n)$. So, the problem $\mathcal{P}5$ can be transformed to

$$
\mathcal{P}6: \quad \min_{\boldsymbol{\alpha}, \boldsymbol{\delta}} E = E' + \lambda \sum_{n=1}^{N} \sum_{m=1}^{M} (|(\alpha_{n,m} - \check{\alpha}_{n,m})|^2 + |\alpha_{n,m}(1 - \check{\alpha}_{n,m})|^2)
\tag{41a}
$$

$$
\text{s.t.} \quad (21g), (21h), (21k), (38b), (40)
\tag{41b}
$$

This is a standard convex optimization problem that can be solved by the CVX toolbox. Throughout each iteration, solving problem $\mathcal{P}6$ is equivalent to solving problem $\mathcal{P}2$. However, it is crucial to note that the solution obtained for $\mathcal{P}6$ may not adhere to the constraints set by $\mathcal{P}2$, as $\mathcal{P}2$ specifically requires $\boldsymbol{\alpha}$ to be integers. Therefore, it is imperative to continuously iterate and solve $\mathcal{P}6$ until the $\boldsymbol{\alpha}$ values converge to integers, signifying the completion of the solution for problem $\mathcal{P}2$. To solve this problem, we need to update the variables $\check{\boldsymbol{\alpha}}$ in each iteration according to the association strategy $\boldsymbol{\alpha}'$ of the previous round, and the closed form of $\check{\boldsymbol{\alpha}}$ can be expressed as [30]

$$
\check{\alpha}_{n,m} = \frac{\alpha'_{n,m} + (\alpha'_{n,m})^2}{1 + (\alpha'_{n,m})^2}, \forall n \in \mathcal{N}, m \in \mathcal{M}
\tag{42}
$$

Based on the above discussion, we can summarize the iterative algorithm as Algorithm 1. In each iteration, the first step is to obtain new power control and computing resource allocation strategies based on the previous round's resource allocation and task-offloading strategies. Next, based on the new power control and computing resource allocation strategies, as well as the previous round's task-offloading strategy, a new task-offloading strategy is obtained by solving $\mathcal{P}6$. Then, the penalty coefficient $\lambda$, i.e., $\lambda = \mu\lambda$, is updated. Finally, the iteration stops when the weighted system energy consumption of the current iteration and the previous iteration does not exceed the maximum tolerance value $\epsilon$.

---

**Algorithm 1:** Joint Task-Offloading and Resource Allocation Algorithm for solving $\mathcal{OP}$

---

**1: Input**: maximum tolerance $\epsilon$, constant parameter $\mu$, where $\mu > 1$, the maximum number of iterations $Iter_{max}$, initial feasible point $\{P_0, t_0, \alpha_0, \delta_0\}$.

**2: for** $i = 1$ to $Iter_{max}$ **do**

**3:**　Update the communication and computation resource allocation strategy $\{P_i, t_i, F_i\}$ by solving $\mathcal{P}3$ based on $\{P_{i-1}, t_{i-1}, \alpha_{i-1}, \delta_{i-1}\}$.

**4:**　Update variables $\dot{\alpha}_i$ with fixed variables $\alpha_{i-1}$ based on (42)

**5:**　Obtain optimal $\{\alpha_i, \delta_i\}$ by solving $\mathcal{P}6$ with given $\{F_i, t_i, \alpha_{i-1}, \delta_{i-1}, \dot{\alpha}_i\}$.

**6:**　Update penalizing coefficient $\lambda$ by $\lambda = \mu\lambda$

**7:**　**if** $\frac{|E_i - E_{i-1}|}{E_{i-1}} \leq \epsilon$

**8:**　　break.

**9:**　**end if**

**10: end for**

**11: Output**: The optimal policy $\{P_i, F_i, \alpha_i, \delta_i\}$ and optimal energy system $E_i$

---

### 5.3. Complexity Analysis

In each iteration of Algorithm 1, the computational complexity is determined by the computational complexities of $\mathcal{P}3$ and $\mathcal{P}6$. The number of optimization variables in problem $\mathcal{P}3$ is $I_1 = MN + M + 4N$, and the number of constraints is $I_2 = MN + 2M + 6N + 1$. We solve $\mathcal{P}3$ using the interior point method; according to [32,33], the computational complexity is $O((I_1^2 I_2 + I_1^3)I_2^{0.5})$. The number of optimization variables in problem $\mathcal{P}6$ is $I_3 = MN + N$, and the number of constraints is $I_4 = 2MN + 3N$. Similarly, the computational complexity for $\mathcal{P}6$ is $O((I_3^2 I_4 + I_3^3)I_4^{0.5})$. Therefore, the computational complexity of the proposed algorithm can be expressed as $O((I_1^2 I_2 + I_1^3)I_2^{0.5}(I_3^2 I_4 + I_3^3)I_4^{0.5}L)$.

## 6. Numerical Result

In this section, numerical simulation results are provided to demonstrate the performance of the proposed algorithm. We consider a square area of 2000 m × 2000 m; the HAP drone is located in the center of this area with an altitude of 20 km [34]. In this system, multiple LEO satellites are randomly distributed at an altitude of 200 km, and GDs are randomly distributed on the ground in this area. We use the MATLAB R2020b (version 9.9.0.1467703) to simulate, and the cvx toolbox used is also installed on MATLAB. Simulation results are obtained on the PC with the Intel Core i5-10505 CPU, 16G RAM, and a 64-bit operating system x64-based processor.

In the proposed system, the NOMA communication scheme is adopted between GDs and the HAP drone, and the FDMA communication scheme is adopted between the HAP drone and LEO satellites. The HAP drone communicates with GDs using 5 GHz bands on the C-band and adopts 31 GHz bands on Ka-band to communicate with LEO satellites. The communication bandwidth between the GD and HAP drones and between the HAP drone and LEO satellites is 100 MHz [9]. AWGN spectral density is $-174$ dBm/Hz [35]. All GDs have the same maximum transmission power of 23 dBm [36], and the maximum transmit power of the HAP drone is 43 dBm [37]. To simplify the experiment, we assume that the delay constraints of all tasks are the same, which are 3 s, and the CPU cycles to compute one-bit tasks are also the same (unless otherwise specified, it is 1000 cycle/bit [38]). The computing capacity of the HAP drone is $F_h = 10$ GHz. We consider that the HAP drone has a stronger computing capacity than a single LEO satellite, so we set the computing capacity of each LEO satellite to be $F_m = 2$ GHz. We set $\omega_g = 1$, $\omega_h = 0.5$, and $\omega_s = 0.2$.

For comparison, the following task-offloading and resource allocation algorithms are employed in the simulations: (i) Pure HAP: all tasks are executed at the HAP drone, and computing and communication resource allocation is obtained by solving $\mathcal{P}3$ (that is $\{\alpha_{n,m} = 0, \forall n \in \mathcal{N}, m \in \mathcal{M}\}$ and $\{\delta_n = D_n, \forall n \in \mathcal{N}\}$). (ii) OMA: the communication scheme between the GD and HAP drones is FDMA. Computing and communication

resource optimization are solved by the convex optimization algorithm, and the task-offloading strategy is obtained by solving $\mathcal{P}6$. (iii) Proposed: the optimization algorithm proposed in this paper.

Figure 2 verifies the convergence of the proposed algorithm in this paper. We plot two curves for the number of ground devices 20 and 40. From Figure 2, we found that the proposed iterative algorithm can quickly converge to a stable solution, and this verifies the convergence properties of our proposed algorithm. So, the algorithm we proposed is an effective algorithm with rapid convergence. The proposed algorithm is based on a single time slot, during which the LEO satellite is assumed to be quasi-static. When considering satellite movement, we can update resource allocation and task scheduling strategies according to multi-dimensional resource information of different time slots.

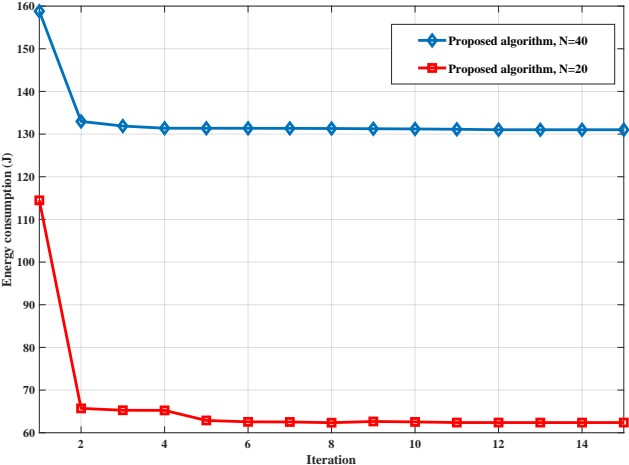

**Figure 2.** Convergence process of proposed algorithm under a different number of ground devices.

Figure 3 shows the sum of weighted energy consumption against the data size of GDs' tasks, in which the time constraints of all GDs are 3 s. The number of LEO satellites is $M = 3$, and the number of GDs is $N = 40$. From Figure 3, we can obtain that as the data size of GDs' tasks increases, the sum weighted energy consumption of the system tends to increase for all schemes because the required energy consumption of the GDs offloading tasks to the edge server of the HAP drone or LEO satellite is positively related to the data size of all tasks. Compared with the two types of benchmark algorithms, the proposed algorithm can ensure the minimum weighted energy consumption, which shows that this algorithm can reduce the weighted energy consumption of the system by optimizing task offloading and resource allocation.

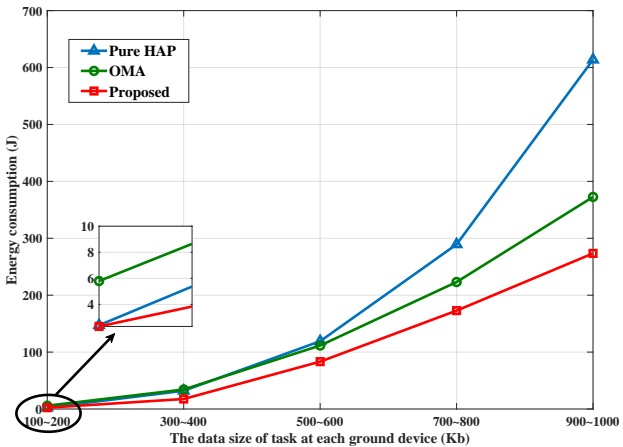

**Figure 3.** Weighted energy consumption of the system versus different task data size.

Figure 4 demonstrates the energy consumption of the system for the three algorithms versus the different number of ground devices. In this figure, we set the number of LEO satellites $M = 3$. The performance is compared at different data sizes of tasks, between 300 Kb and 400 Kb, and between 900 Kb and 1000 Kb. From the figure, we can get that the larger the number of ground devices, the more energy consumption for ground devices to offload their tasks. In this figure, we can also see that the proposed iterative algorithm has much lower energy consumption compared to the two benchmark algorithms. Moreover, the greater the amount of task data, the more the performance of the proposed algorithm will be improved.

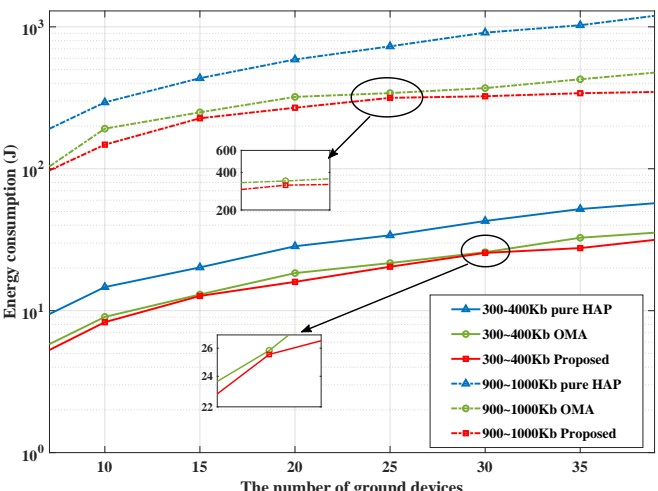

**Figure 4.** Weighted energy consumption of the system versus different number of ground devices.

Figure 5 shows the energy consumption for the three algorithms versus different maximum tolerance delay, where $T = \{2, 2.5, 3, 3.5, 4, 4.5, 5\}$ (s). We set the number of ground devices as 20, and there are three LEO satellites. Based on the analysis of Figure 5, it is evident that the energy consumption of all algorithms decreases as the maximum tolerance delay increases. This is because with the increasing of the maximum tolerance delay, the transmit power of GDs and HAP drone can be smaller, and the CPU resource allocated to all tasks can also be smaller, which results in lower system energy consumption. By comparing the energy consumption of the three algorithms, we can get that the proposed algorithm can obtain smaller system energy consumption, and the smaller the tolerance delay of tasks, the more obvious the ability of the proposed algorithm to reduce energy consumption.

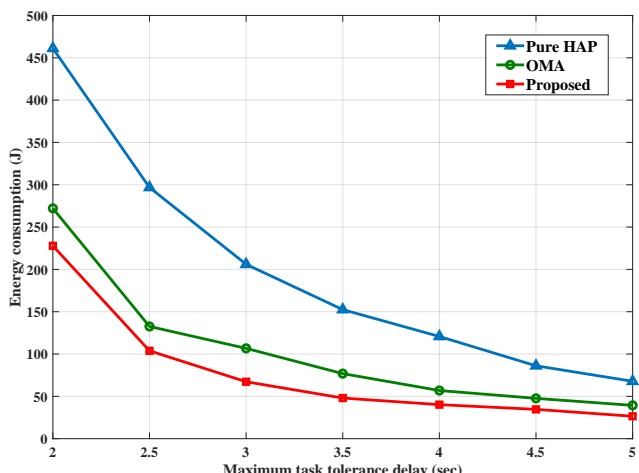

**Figure 5.** Weighted energy consumption of the system versus different maximum tolerance delay.

Figure 6 displays the weighted energy consumption versus different required CPU numbers for one-bit task data. In this figure, we set the number of ground devices as 40, and we set the number of LEO satellites as 3. From Figure 6, we can see that, for a fixed number of ground devices and LEO satellites, the energy consumption increases with the required number of CPU cycles for one-bit data. The energy consumption is consistently lower than the other two benchmark algorithms.

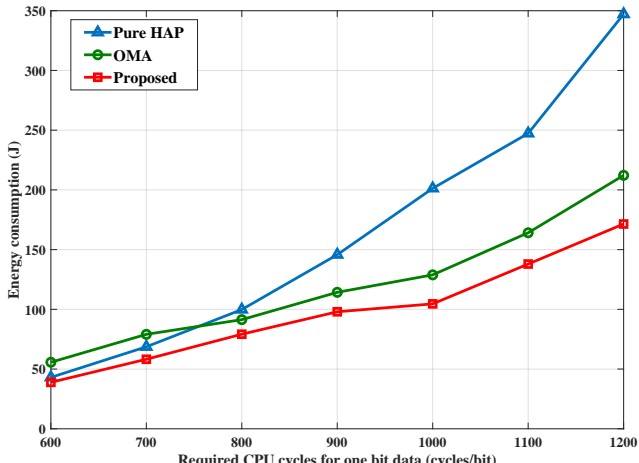

**Figure 6.** Weighted energy consumption of the system versus different CPU cycles for one-bit task data.

Figure 7 depicts the energy consumption of the system for three different algorithms with a different number of LEO satellites. We compared the performance of three algorithms in two scenarios where the number of ground devices is 20 and 40. Compared with the other two algorithms, the energy consumption of the pure HAP algorithm is always the highest and does not change with the number of LEO satellites. This is because the task-offloading process of the pure HAP algorithm does not involve the participation of LEO satellites. The energy consumption of the other two algorithms decreases with the increase in the number of LEO satellites. This is because the weight of satellite energy consumption is lower, and as the number of satellites increases, more task data can be allocated to LEO satellites. The energy consumption of the proposed algorithm is always the lowest, which further illustrates the performance of the proposed algorithm.

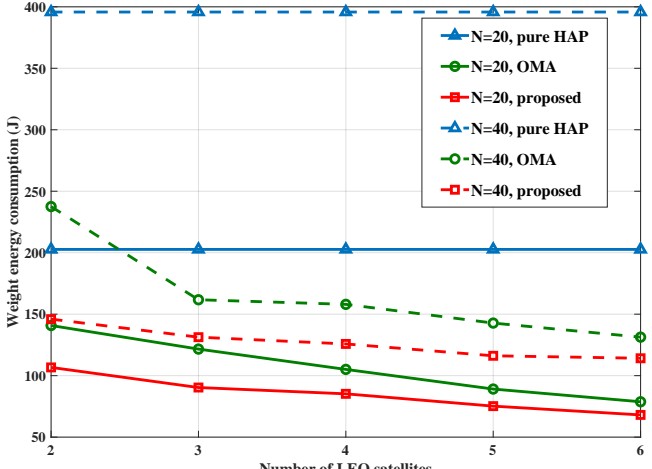

**Figure 7.** Energy consumption of the system versus different number of LEO satellites.

## 7. Conclusions

In this paper, we focus on the space–air–ground collaborative network. Ground devices communicate with the HAP drone based on the NOMA technique. The HAP

drone can process part of a GD's task locally while offloading the rest of the task to LEO satellites for processing. We formulate an optimization problem to jointly optimize multiple resource allocation and task offloading to minimize the weighted energy consumption of the system while ensuring the maximum task tolerance delay is met. We proposed an iterative algorithm that can converge quickly to reduce the complexity of the original problem. The simulation results verify the convergence and performance of the proposed algorithm compared to the other two benchmark algorithms. There are two main research directions in the future. In terms of scenarios, the research on scenarios involving multiple HAP drones and multiple satellites, covering a broader range, will become a new research trend. In terms of algorithms, distributed algorithms such as federated learning will receive more in-depth research.

**Author Contributions:** Conceptualization, C.M. and C.G.; methodology, C.G. and N.J.; validation, N.J., C.G., and B.H.; formal analysis, C.G. and N.J.; investigation, H.W. and Y.X.; writing—original draft preparation, C.G.; writing—review and editing, C.M., H.W., and B.H.; supervision, C.M. and H.W.; project administration, Y.X.; funding acquisition, C.M. All authors have read and agreed to the published version of the manuscript.

**Funding:** This work was supported by the 2020 National Key R&D Program "Broadband Communication and New Network" special "6G Network Architecture and Key Technologies" 2020YFB1806700.

**Data Availability Statement:** Not applicable.

**Conflicts of Interest:** The authors declare no conflicts of interest.

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
