# Peer review of "Joint Task Offloading and Resource Allocation for Space–Air–Ground Collaborative Network"

_drones, doi:10.3390/drones7070482_

Round 1
Reviewer 1 Report
Comments and Suggestions for Authors
1. Authors focused on space-air-ground collaborative network.
2. Authors proposed an algorithm that optimizes joint multiple resource allocation and task offloading strategies to minimize the sum weighted energy consumption.
3. Authors proposed an iterative algorithm to solve the original problem which is a MINLP problem.
4. Paper is well presented and results are very exciting.
Reviewer 2 Report
Comments and Suggestions for Authors
This article investigated weighted energy consumption minimization problem for space-air-ground collaborative network, where a low-complexity iterative algorithm was proposed to optimize power control, computing frequency allocation, and task offloading decision. There are some concerns listed as follows.
1. Algorithm 1 lacks the step of penalizing coefficient update in the iterative process, which will cause integer variables to fail to converge during the running of the algorithm. The authors should add the process of updating the penalty coefficient in Algorithm 1 and provide a detailed explanation of the proposed algorithm's operation in the article.
2. The article lacks a complexity analysis of the proposed algorithm, which leads to the inability to fully explain the efficiency of the algorithm.
3. Please provide an explanation of the location of the control center for information collection, decision-making, and distribution, as well as the specific process involved in the space-air-ground collaborative network scenario presented in the article.
Comments on the Quality of English LanguageSome sentences need to be improved.
Reviewer 3 Report
Comments and Suggestions for Authors
The paper proposes a design of joint task offloading and resource allocation for space-air-ground collaborative networks. The topic of this paper is very interesting and timely. The proposed joint optimization problem of task offloading and resource allocation is solid. The performance evaluations are convincing. The reviewer just has some minor suggestions that may help improve the quality of this paper as follows.
*First, although the topic of this paper is timely, the paper should clarify its key novelties and technical contributions compared to the existing studies. In particular, there have been several related studies about the joint task offloading and resource allocation via NOMA transmission, e.g., "NOMA-Assisted Multi-Access Mobile Edge Computing: A Joint Optimization of Computation Offloading and Time Allocation," IEEE Transactions on Vehicular Technology, vol. 67, no. 12, pp. 12244-12258, Dec. 2018 and "Joint Task Offloading and Resource Allocation for NOMA-Enabled Multi-Access Mobile Edge Computing," IEEE Transactions on Communications, vol. 69, no. 3, pp. 1548-1564, March 2021. Thus, the paper should clarify its key novelties and technical contributions.
*Second, as the LEOs are moving, it is interesting for the paper to discuss how to extend the results in this paper into the scenario of moving LEOs.
*Third, below eq. (16) and eq. (17), “Where” should read “where”.
*Fourth, there are several problem formulations, i.e., P1 to P6, in this paper. How are these problem formulations related to each other? The paper should provide more illustrations.
*Fifth, it is interesting for the paper to discuss how to implement the proposed algorithm, i.e., Algorithm 1, in practical LEO systems.
*Sixth, some future directions of this work could be discussed in the conclusion section of this paper.
